# Galacto-Oligosaccharide (GOS) Synthesis during Enzymatic Lactose-Free Milk Production: State of the Art and Emerging Opportunities

Katia Liburdi * and Marco Esti

Department of Agriculture and Forest Sciences (DAFNE), Tuscia University, Via San Camillo de Lellis, 01100 Viterbo, Italy; esti@unitus.it
* Correspondence: k.liburdi@unitus.it

**Abstract:** Much attention has recently been paid to β-Galactosidases (β-D-galactoside galactohidrolase; EC 3.2.1.23), commonly known as lactases, due to the lactose intolerance of the human population and the importance of dairy products in the human diet. This enzyme, produced by microorganisms, is being used in the dairy industry for hydrolyzing the lactose found in milk to produce lactose-free milk (LFM). Conventionally, β-galactosidases catalyze the hydrolysis of lactose to produce glucose and galactose in LFM; however, they can also catalyze transgalactosylation reactions that produce a wide range of galactooligosaccharides (GOS), which are functional prebiotic molecules that confer health benefits to human health. In this field, different works aims to identify novel microbial sources of β-galactosidase for removing lactose from milk with the relative GOS production. Lactase extracted from thermophilic microorganisms seems to be more suitable for the transgalactosylation process at relatively high temperatures, as it inhibits microbial contamination. Different immobilization methods, such as adsorption, covalent attachment, chemical aggregation, entrapment and micro-encapsulation, have been used to synthesize lactose-derived oligosaccharides with immobilized β-galactosidases. In this mini-review, particular emphasis has been given to the immobilization techniques and bioreactor configurations developed for GOS synthesis in milk, in order to provide a more detailed overview of the biocatalytic production of milk oligosaccharides at industrial level.

**Keywords:** lactose; β-galactosidase; transgalactosylation; galacto-oligosaccharides (GOS)

## 1. Introduction

Lactose is a disaccharide composed of two aldohexoses, chemically defined as O-β-D-galactopyranosyl-(1-4)-β-D-glucose, which is soluble in water (170 g/L at 15 °C) and six times less sweet than sucrose [1]. As lactose is the primary carbohydrate source found in most mammalian milks, it is essential for the development of neonatal mammals, due to the appropriate balance of glucose and galactose provided by its molecule. Although glucose is considered to be a primary energy source, galactose is known to play a crucial role in mammalian brain development and has important nutritional and prebiotic properties [2–4]. As occurs with other sugars, lactose is hydrolyzed into its monosaccharide components, glucose and galactose, by the lactase-phlorizin-hydrolase enzyme complex (LPH), commonly known as lactase produced in the intestinal lumen (Figure 1). The mature human LPH is an integral glycoprotein that has four homologous structural domains involved in the intramolecular organization of the enzyme. LPH belongs to a group of intestinal glycoside hydrolases with two strongly associated enzymatic activities with partly independent catalytic sites: lactase (β-D-galactoside galactohydrolase, EC 3.2.1.23), which is responsible for lactose hydrolysis, and phlorizin hydrolase (glycosyl-N-acylsphingosine glucohydrolase, EC 3.2.1.62), which is required for B-glycosylceramide digestion [5,6]. These two main catalytic activities are reflected in the term "lactase phlorizin hydrolase", the aforementioned full name of the enzyme.

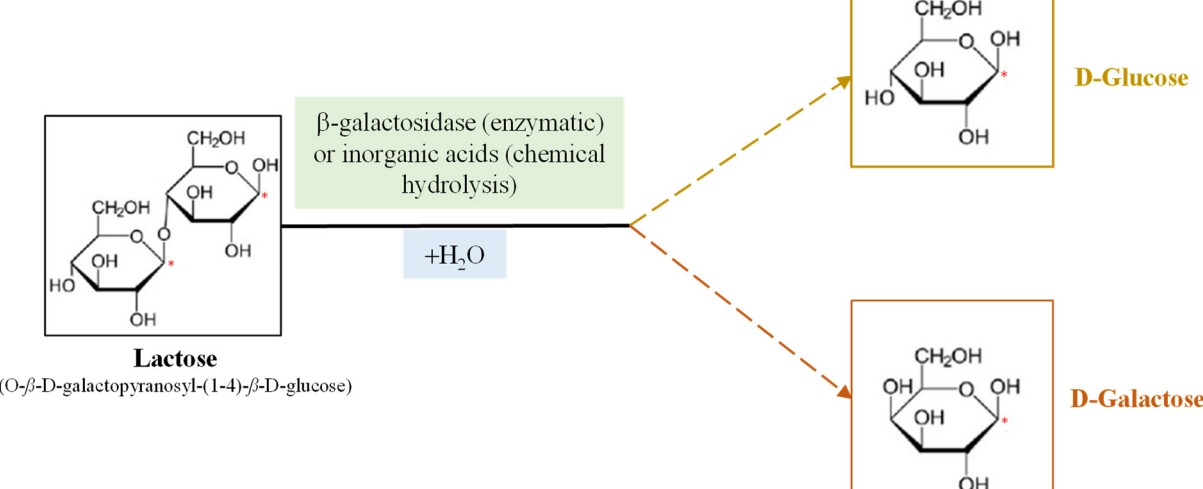

**Figure 1.** Graphical illustration of lactose hydrolysis. * indicates anomeric groups.

LPH is the most important glycosidase in post-natal mammalian life, since lactose is the main carbohydrate ingested during this period. Subsequently, lactase expression decreases as the organism grows older and the significance of lactose in daily nutrient ingestion diminishes [7].

Only a few humans continue to produce lactase beyond weaning, which is known as lactase persistence (LP). It is commonly found in Northern European populations, probably due to their recurring consumption of milk and dairy products as part of their daily diet [8]. However, in other geographical areas, where the regular intake of milk stops after breastfeeding is over. A total of 70% of the world's population is lactase deficient, which is known as lactose intolerance (LI) [9], defined as the pathophysiological situation in which the small intestinal digestion and/or colonic fermentation is altered, which induces clinical symptoms. The dose of lactose that will cause symptoms differs among individuals, depending on the quantity of lactose consumed, the degree of the lactase deficiency and the form of food in which the lactose is ingested [8].

Lactose-free dairy (LFD) products can provide essential nutrients present in milk to lactose-intolerant people. There is a growing inclination of consumers towards LFD products as they are becoming more health-conscious and plant-based, non-dairy alternatives are available for individuals with lactose intolerance, especially in countries with the highest prevalence of LI [10].

The LFD market is the fastest growing segment in the dairy industry. It is expected to reach a USD 9 billion turnover by 2022 and continues to overtake the traditional dairy market (7.3% vs. 2.3%) [11]. As reported by the Euromonitor database (2018), the global lactose-free food market will continue to grow at a compound annual growth rate (CAGR) of 6% over the 2015–2020 forecast period, reaching USD 8.8 billion in 2020. Lactose-free dairy, which is expected to reach a slightly higher CAGR of 7%, will account for 80% of this. The largest category of lactose-free dairy products is lactose-free milk (LFM), which represents two-thirds of the market and drives the absolute growth (Figure 2).

The microbial β-galactosidase, commonly known as lactase, is primarily used to produce LFD products at the industrial scale, although the glucose and galactose produced by lactose hydrolysis make the milk much sweeter. However, various studies conducted on the sensory characteristics of LFM milk compared with traditional milk revealed that lactose-free milk tastes sweeter than regular milk [12–14], due to the fact that lactose enzymatic hydrolysis produces glucose and galactose, which are sweeter than lactose.

Based on the results obtained by Harju [12], it is clear that differences exist between lactose-free milk and regular milk, not only in terms of sweetness, but also due to the perception of chalkiness and higher viscosity in lactose-free milk. Moreover, the presence

of monosaccharides produced during lactose hydrolysis in LFD products might increase the risks for certain groups of people such as diabetics [13].

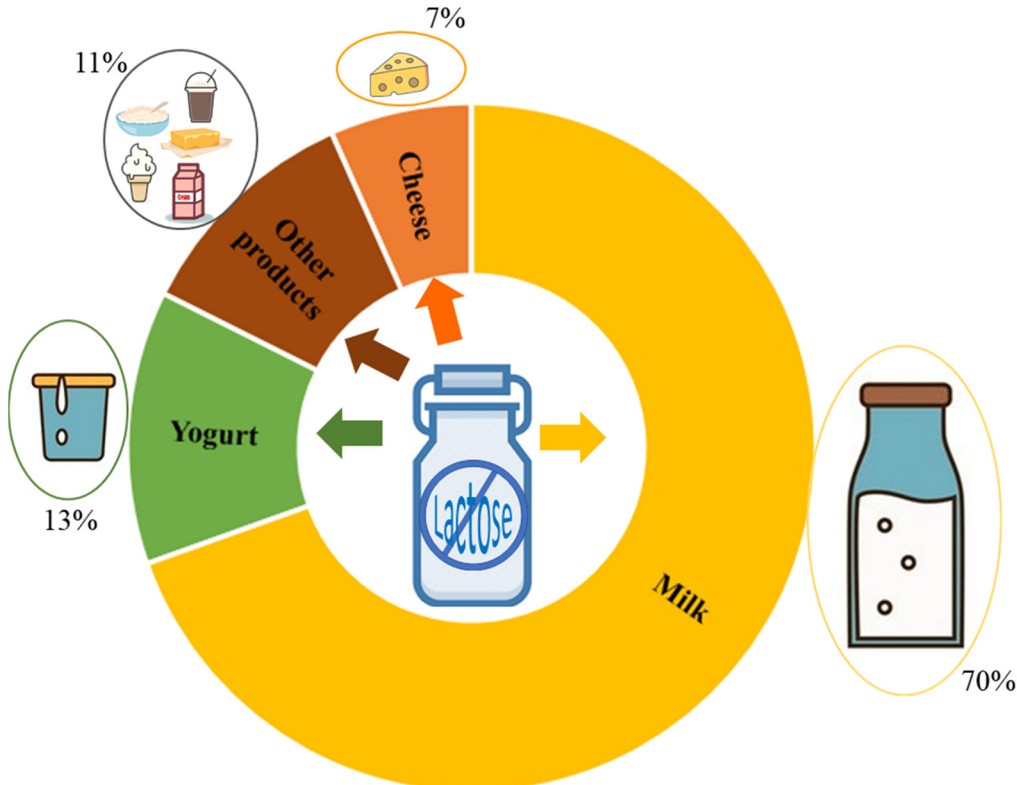

**Figure 2.** Target market identified for LFD products. The figure was elaborated using the data published by Dekker et al. [11].

In addition to lactose hydrolysis, $\beta$-galactosidases are able to catalyze a transgalactosylation reaction in which lactose in the mixture serve as galactosyl acceptors, yielding galacto-oligosaccharides (GOS) [15–17]. These are non-digestible oligosaccharides and glycosides with prebiotic activity and other functionalities, increasingly used as ingredients in functional foods and some pharmaceuticals.

The $\beta$-galactosidases source and the initial lactose concentration influence the yield and composition of the synthesized GOS [18]. Of the various microbial sources, the $\beta$-galactosidase from mesophilic yeast *Kluyveromyces* spp. has been widely used in the dairy process for its exceptional lactose hydrolysis activity. However, this enzyme has some weaknesses due to its low transglycosylation and thermostability activity for GOS production in LFM [19,20]. On the other hand, the thermostable $\beta$-galactosidases exhibits higher transglycosylation yields due to their higher reaction rate and long half-lives at temperatures in which lactose is more soluble [21,22]. In this respect, these enzymes result a successful alternative to mesophilic $\beta$-galactosidases for the industrial processing of lactose-free dairy products.

This mini-review focuses on recent advances in GOS production from lactose in LFM products.

Recent studies on reaction mechanisms, structure and sources of β-galactosidases, and the factors affecting GOS yields will be compared. Lastly, β-galactosidases immobilization techniques and bioreactor configurations for GOS synthesis in milk will be discussed. More specifically, the aim of this review is to provide knowledge related to the study on biocatalytic synthesis of galacto-oligosaccharides in milk medium.

## 2. Enzyme Sources and Factors Affecting GOS Production

As reported in Figure 3, *β*-Galactosidase possesses hydrolytic and transgalactosylation activities [23,24].

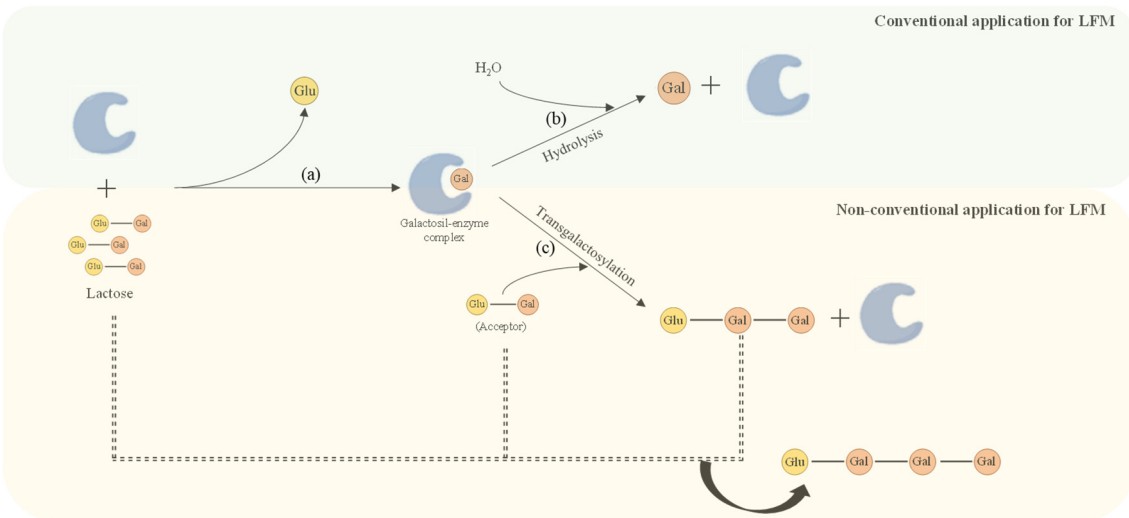

**Figure 3.** Catalytic mechanism of the *β*-galactosidase enzyme with lactose as substrate: (**a**) enzyme–galactosyl complex formation; (**b**) hydrolytic reaction; (**c**) transgalactosylation mechanism. Glu: Glucose; Gal: Galactose; LFM: Lactose-free Milk.

In the hydrolysis reactions catalyzed by β-galactosidase, the formation of an enzyme–galactosyl complex occurs upon a simultaneous release of glucose (Figure 3a) and a transfer of the enzyme–galactosyl complex to an acceptor that contains a hydroxyl group [25]. A hydrolytic reaction forms glucose and galactose from lactose, and galactose is obtained as a product if the acceptor is the water [25,26] (Figure 3b).

Otherwise, the lactose in the medium can also operate as an acceptor, and, under these occurrences, GOS are formed instead through the transgalactosylation mechanism (Figure 3c), excluding a water molecule under controlled conditions.

Regarding the transgalactosylation activity of *β*-galactosidase, it is not clear whether the interaction in the active site depends on the acceptor species which can be a saccharide or water. However, different enzymes have been found to have different affinities for water and saccharides, as different enzymes yield different amounts of GOS at the same lactose concentration [27,28]. It seems likely that the enzyme source is the main factor that profoundly influences the reactions of hydrolysis and transgalactosylation [29].

*β*-galactosidases are widely distributed in numerous biological systems, e.g., microorganisms, plants and animal tissues; however, compared to animal and plant sources, microorganisms produce higher yields of enzymes, resulting in a decline in the prices of commercial preparation [30].

In a recent publication, Fisher [18] gave a full overview of the obtained GOS yields by various studies, sorted by type of medium and enzyme source. This study reveals that different media can be considered suitable substrates for GOS synthesis, but reaction parameters, especially enzyme origin, have to be selected accurately.

It is well known that in this context, the catalytic properties *β*-galactosidase function and specificity differ significantly on the microbial source, in terms of molecular weight, amino-acids chain length, position of the active site, pH and thermal optimum and stability [31].

Of the various microorganisms, the *β*-galactosidase from the mesophilic yeast *K. lactis* has been mainly used in LFD production due to its dairy environmental habitat and remarkable lactose hydrolysis efficiency [20]. However, *K. lactis* *β*-galactosidase has some disadvantages due to its low transgalactosylation activity and poor thermostability properties.

It is advisable to use thermophilic microorganisms that are able to produce GOS at high temperatures because it enables better control of the solubility of lactose as a substrate and inhibits contamination. Moreover, the higher temperature enhances the lactose solubility, and hence, high initial concentrations saccharide of can enhance GOS synthesis. [16,17,20]. Therefore, thermostable $\beta$-galactosidases obtained from thermophilic and hyperthermophilic microorganisms have recently attracted considerable attention.

As reported in Table 1, the microbial $\beta$-galactosidases with transgalactosylation activity were frequently isolated from mesophiles and different categories of thermophiles. From the data reported in Table 1, it emerges that yields above 50% are seldom overcome. More customary optimized yields are within 30% and 40% (*w/w*). The maximum GOS yield of $\beta$-galactosidases produced from the moderate thermophiles *Bullera singularis* is recorded at 37 °C [32], while *Sulfolobus solfataricus* incubated at 70 °C produced the highest GOS yield of $\beta$-galactosidases for the extremophile and hyperthermophile categories [33]. Therefore, temperature appears to affect GOS synthesis in the medium. Several studies have found that higher temperatures give higher GOS yields [34,35]. However, single enzymes respond to temperature in different ways as various studies have revealed that GOS yield remained constant at different temperatures [36–38].

**Table 1.** The optimum of temperature ($T_{opt}$, °C) and pH ($pH_{opt}$) of microbial $\beta$-galactosidases with transgalactosylation activities.

|  | Microrganisms | $T_{opt}$ (°C) | $pH_{opt}$ | Maximum GOS Yield (%, *w/w*) | References |
|---|---|---|---|---|---|
| Hyperthermophiles | *Thermotoga maritima* | 80–85 | 6.5 | 19 | [39] |
| Extreme thermofiphiles | *Sulfolobus solfataricus* | 80 | 6.5 | 41 | [40] |
|  | *Thermus aquaticus* | 80 | 5.5 | 39 | [41] |
|  | *Pyrococcus furiosus* | 80 | 5.0 | 33 | [33] |
| Moderate thermophiles | *Saccharopolyspora rectivirgula* | 70 | 7.0 | 44 | [40] |
|  | *Sterigmatomyces elviae* | 60 | 5.0 | 39 | [42] |
|  | *Geobacillus stearothermophilus* | 70 | 7.7 | 18.6 | [43] |
|  | *Lactobacillus acidophilus* | 55 | 6.5–8 | 25.5 | [44] |
|  | *Bullera singularis* | 50 | 6.0 | 50 | [32] |
|  | *Lactobacillus reuteri* | 50 | 6.5–8 | 38 | [36] |
|  | *Bifidobacterium longum* | 45 | 6.8 | 32.5 | [34] |
| Mesophiles | *Aspergillus oryzae* | 40 | 4.5 | 27 | [45] |
|  | *Kluyveromyces lactis* | 40 | 7.0 | 25 | [46] |
|  | *Bifidobacterium bifidum* | 37 | 6.5 | 20 | [47] |
|  | *Kluyveromyces marxianus* | 30 | 6.5 | 25 | [48] |

Moreover, it is known that pH can influence the kinetics of lactose hydrolysis and the related GOS production [49], thus suggesting that it may be helpful to control the synthesis and degradation rates of oligosaccharides by varying the pH of the medium, thus increasing the GOS yields. The altered optimum pH affected the ionization state of the catalytic residue to favor interaction with the sugar acceptor and resulted in an improved ratio of transglycosylation [50]. However, this property may vary between single enzymes, such as the effect of temperature on yield [17].

Another important factor regarding enzyme source is the relationship between the GOS yield and the lactose conversion (percentage of initial lactose that is consumed during the synthesis), as it has very important nutritional and technological consequences. As previously mentioned, it is advisable to reduce lactose concentration in lactose-free milk products. For example, $\beta$-galactosidase from *Aspergillus oryzae* yields 27% of GOS with a lactose conversion of 58% [45]. However, unfortunately, this mesophilic microorganism is

unable to produce GOS at high temperatures where better lactose solubility and reduced microbial contamination could be assured. The effects of the initial lactose concentration are linked to two fundamental factors: (i) the greater availability of galactosyl acceptor saccharides and (ii) the reduced availability of water. The former (i) should increase the rate of GOS synthesis, and the latter (ii) should reduce the rates of both GOS degradation and hydrolysis of lactose [17].

Moreover, several kinetic studies reveal that the main drawback of using β-galactosidases for GOS production is difficulty achieving complete lactose hydrolysis because the end-products galactose and glucose inhibit the enzyme activity [20,51]. At low lactose concentrations, galactose inhibition is more effective than glucose. In view of the foregoing, it is not surprising that an enzyme with significantly low inhibition of lactose hydrolysis by glucose is advantageous [20].

## 3. Recent Advances in β-Galactosidase Immobilization for GOS Production

Due to the great interest in using lactase for GOS production, immobilized enzyme systems are being intensively investigated for possible industrial enzyme applications. As previously mentioned, it is essential to note that the yield and selectivity of GOS synthesis are strongly dependent on the enzyme source [52]. Moreover, immobilization has been reported to influence the catalytic efficiency of enzymes (pH and temperature profiles, stability and kinetic parameters) and may subsequently change the affinity and reactivity for the saccharide donor [53–56]. Therefore, a well-considered selection of the immobilization procedure may enhance the catalytic properties of the enzyme for a given target product.

Enzyme immobilization is known to have many advantages: it enhances the operational stability of the enzyme, provides better operational control and allows higher flexibility of reactor design, easy product recovery as well as easy catalyst recovery and reuse [56]. Several immobilization techniques can be divided into classes that involve different enzyme-support interactions (chemical, physical or physico-chemical).

β-Galactosidase has been immobilized by different methods such as physical adsorption, gel entrapment and covalent attachment on various carriers, and several immobilized enzyme systems have been studied for hydrolysis of milk lactose [57]. Conventionally, immobilized β-galactosidase is added directly to whole milk during the large-scale industrial lactose hydrolysis, which is cost-effective due to the possibility of reusing the enzyme after lactose hydrolysis is complete. The Centrale del Latte of Milan, Italy, employed SNAM Progetti technology that used *K. lactis* β-galactosidase entrapped into cellulose triacetate fibers as catalyst for lactose hydrolysis [58]. Although the lactose hydrolysis and transgalactosylation with β-galactosidase from *A. oryzae* are well documented at the industrial level, novel and efficient immobilized preparations are still required.

In recent publications, β-Galactosidase from A. oryzae has been immobilized in a broad range of materials, mainly by covalent attachment, to produce biocatalysts with transgalactosylation activity. Different functionalized mesoporous supports, biopolymers and nanoparticles have been employed. A commercial preparation of lactase extracted from *A. oryzae* was covalently immobilized [59] by using amino-glyoxyl-agarose as support, the maximum GOS obtained was approximately 5% [55]. However, the GOS yield by transgalactosylation activity was 20% when the same β-galactosidase was immobilized on glyoxyl-functionalized porous silica support, whereas the yield of the free enzyme form was only 11% [60]. These results suggest that the immobilization method results in a positive impact on lactose transgalactosylation if the chemical binding did not damage the active site.

However, some authors suggest that oriented covalent immobilization may affect biocatalyst activity and stability, and therefore, its final biocatalyst performance. Since proteins have some areas that are more prone to unfolding on their surfaces or in proximity [61], binding through these areas may improve final enzyme stability. One technique for modifying enzyme orientation is to use a two-step immobilization process on heterofunctional

supports [62–64]. Generally, the heterofunctional support may be defined as that which possesses several distinct functionalities on its surface able to interact with an enzymatic protein. Urritia et al. [65] studied the use of chitosan heterofunctionality for the covalent immobilization of *A. oryzae β*-galactosidase in a two-step process. The GOS synthesized by immobilized enzyme were over 20% ($w/w$), and after 10 sequential batches, the cumulative GOS productivity obtained with the chitosan biocatalysts were 4.7 times more than when soluble *β*-galactosidase was used.

Due to the structure and distribution of residual amino acid of *A. oryzae β*-galactosidase, protein immobilization by ionic interaction is possible [59]. The essential advantages of this fast and easy immobilization procedure are that no additional reagents or modifications of the enzyme are needed. Therefore, the protein structure is severely affected by ionic binding so that immobilization yields are likely to be high, and enzyme-support interaction is mildly sufficient to allow for support recovery after exhaustion of enzymatic activity, thus reducing costs. *A. oryzae* lactase was immobilized by the ionic mechanism in a quaternary ammonium agarose support. The biocatalyst obtained showed the highest GOS yield and specific productivity values of 24 % ($w/w$) and 9.78 g·g$^{-1}$ h$^{-1}$, respectively [59]. In this study, the cumulative weight of generated oligosaccharides was higher under repeated batch mode with immobilized *β*-galactosidase than the value obtained with a soluble enzyme under single batch mode.

Based on this background information, the covalent binding is often employed to increase the stability and reusability of the enzymes. However, this provided stabilization is usually counterbalanced by partial enzyme deactivation. This negative effect can be mitigated by carefully optimizing the immobilization and reaction conditions. Moreover, immobilized *β*-galactosidases are reusable, and it can be applied in continuous processes with easier downstream operations (i.e., biocatalyst recovery).

Immobilized enzymes enable the optimum utilization of their activity in a bioreactor, thus improving process efficiency. GOS production using immobilized lactase in a continuous packed bed reactor (PBR) could be an effective alternative for GOS production in milk. Warmerdam et al. [66] reported that the product composition of GOS obtained using immobilized lactase on Eupergit® in a PBR was similar to the GOS composition obtained with free lactase in a batch reactor. The enzymatic productivity of immobilized enzymes during one run in the PBR is more than six-fold higher than the productivity of free enzymes during one run in a batch reactor. However, in the same study, immobilized lactase in PBR showed a slightly higher hydrolysis rate (galactose production) and a slightly lower GOS production rate than those obtained with free enzyme in a batch reactor, which is probably due to the diffusion-limiting condition inside the PBR. Subsequently, Carević et al. [67] reported on the GOS production from *A. oryzae β*-galactosidase immobilized on microporous carrier (Purolite® A-109) tested in a fluidized bed bioreactor (FBR). The lactase modified in FBR exhibited a significantly higher stability than the free enzyme, and retained approximately 75% of its activity after 10 cycles in an FBR. These results may be due to the reduced protein structure mobility after the covalent immobilization since more rigid forms tend to be less exposed to damaging environmental effects [68]. In this study, the immobilized biocatalyst was found to have a higher affinity toward catalyzing transgalactosylation than hydrolysis reaction. It is feasible that transgalactosylation is favored by the *β*-galactosidase immobilization due to the hydrophobic nature of the carrier surface, which leads to a lower water concentration in the enzyme microenvironment than occurs in the medium with the free enzyme [67].

The optimal performance obtained with the FBR was predictable as several studies have reported on the widespread and successful application of FBRs in the food industry, which generates benefits from the continuous operational mode as well as the improved mass transfer [69], which may be important for the enzyme bioreactor application in a heterogeneous mixture such as milk.

## 4. Conclusions and Perspectives

Due to the key role of galacto-oligosaccharides in the field of functional foods, in the last few years, transgalactosylation activity of $\beta$-galactosidases has attracted the attention of many researchers. Active research is ongoing to find newly identified microbial sources of $\beta$-galactosidase for removing lactose from milk with the relative GOS production. Using enzymes extracted from thermophilic microorganisms proved to be effective, as increased lactose solubility at higher temperatures generally increases the GOS yield in the medium. End-product inhibition by galactose is another aspect to consider when selecting the β-galactosidase source.

Galacto-oligosaccharides production from microbial lactase and enzyme immobilization has been studied to make the process economically feasible by improving GOS yields and productivity. The transgalactosylation efficiency of the immobilized biocatalysts was generally successfully optimized in packed and fluidized bed bioreactors, although a better performance was obtained with the latter due to its higher mass transfer efficiency.

This review has highlighted important recent findings that have given us a better understanding of $\beta$-galactosidase activity and structure, i.e., the use of immobilized biocatalyst for the hydrolysis of lactose with transgalactosylation, which is a topic of considerable scientific and technological interest and can contribute to the expansion of an efficient GOS production method for application in the dairy industry.

Thus, thermostable enzymes will have great potential in lactose hydrolysis and will be of particular interest to researchers; simultaneously, the $\beta$-galactosidases immobilization techniques will also be an area of great interest. Together, these aspects can help decrease the GOS cost production and increase the prebiotic value of lactose-free milk.

**Author Contributions:** Conceptualization, K.L.; Writing–Original Draft Preparation, K.L.; Writing–Review & Editing, K.L. and M.E.; Supervision, M.E. All authors have read and agreed to the published version of the manuscript.

**Funding:** This research received no external funding.

**Conflicts of Interest:** The authors declare no conflict of interest.

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
