# Peer review of "Galacto-Oligosaccharide (GOS) Synthesis during Enzymatic Lactose-Free Milk Production: State of the Art and Emerging Opportunities"

_beverages, doi:10.3390/beverages8020021_

Round 1
Reviewer 1 Report
The authors show an interesting mini-review on recent advances in the production of GOS from lactose in LFM products.
The manuscript is written coherently and neatly, making it easy to understand.
The importance of GOS and its relationship with beta galactosidases from extremophilic organisms should be more highlighted in the introduction. What is the problem of the current industry? It should be made more explicit.
Towards the end of the review, the authors should discuss their own conclusions. What is the research or industrial route to develop beta galactosidases from extremophilic organisms for GOS production? After this review, what next to answer that problem? Only search in extremophilic organisms? from which environment? with the same techniques that have always been searched? How to take a step towards innovation?
There are few references for the last five years. The authors should update with more recent work if available.
Author Response
Dear reviewer,
many thanks for your suggestions.
Below you can find the response to your questions.
The corrections in the revised manuscript are made in red font.
Best regards
Katia Liburdi
Comments (C) and response (R)
C: The authors show an interesting mini-review on recent advances in the production of GOS from lactose in LFM products.
The manuscript is written coherently and neatly, making it easy to understand.
R: we thank you for your positive comments.
C: The importance of GOS and its relationship with beta galactosidases from extremophilic organisms should be more highlighted in the introduction. What is the problem of the current industry? It should be made more explicit.
R: as suggested by the reviewer, the relationship with thermostable b-galactosidases and GOS production is now discussed on lines 97-107
C: Towards the end of the review, the authors should discuss their own conclusions. What is the research or industrial route to develop beta galactosidases from extremophilic organisms for GOS production? After this review, what next to answer that problem? Only search in extremophilic organisms? from which environment? with the same techniques that have always been searched? How to take a step towards innovation?
R: I’ve added a brief discussion about the reviewer’s questions in the last lines of the “Conclusions” (lines 311-315).
C: There are few references for the last five years. The authors should update with more recent work if available.
R: more recent references are now reported:
Abdul Manas, N. H.; Md. Illias, R.; Mahadi, N. M. Strategy in manipulating transglycosylation activity of glycosyl hydrolase for oligosaccharide production. Crit. Rev. Biotech. 2018, 38(2), 272-293
de Albuquerque, T. L.; de Sousa, M.; Silva, N. C. G.; Neto, C. A. C. G.; Gonçalves, L. R. B.; Fernandez-Lafuente, R.; Rocha, M. V. P. β-Galactosidase from Kluyveromyces lactis: Characterization, production, immobilization and applications-A review. Int. J. Biol. Macromol. 2021, 191, 881-898.
Fischer, C., & Kleinschmidt, T. (2018). Synthesis of galactooligosaccharides in milk and whey: a review. Comp. Rev. Food Sci. Safe. 2018, 17(3), 678-697.
Huang, J.; Zhu, S.; Zhao, L.; Chen, L.; Du, M.; Zhang, C.; Yang, S. T. A novel β-galactosidase from Klebsiella oxytoca ZJUH1705 for efficient production of galacto-oligosaccharides from lactose. App. Microb.Biotech. 2020, 104(14), 6161-6172.
Jenab, E.; Omidghane, M.; Mussone, P.; Armada, D. H.; Cartmell, J.; Montemagno, C. Enzymatic conversion of lactose into galacto-oligosaccharides: The effect of process parameters, kinetics, foam architecture, and product characterization. J. Food Eng. 2018, 222, 63-72.
Movahedpour, A.; Ahmadi, N.; Ghalamfarsa, F.; Ghesmati, Z.; Khalifeh, M.; Maleksabet, A.; ... et al. Savardashtaki, A. (). β‐Galactosidase: From its source and applications to its recombinant form. Biotech. Appl. Biochem. 2021
Neto, C. A. C. G.; Silva, N. C. G.; de Oliveira Costa, T.; de Albuquerque, T. L.; Gonçalves, L. R. B.; Fernandez-Lafuente, R.; Ro-cha, M. V. P. The β-galactosidase immobilization protocol determines its performance as catalysts in the kinetically con-trolled synthesis of lactulose. Int. J. Biol. Macromol. 2021, 176, 468-478.
Park, A.-R.; Oh, D.-K. Galacto-oligosaccharide production using microbial β-galactosidase: current state and
Ricardi, N. C.; Arenas, L. T.; Benvenutti, E. V.; Hinrichs, R.; Flores, E. E. E.; Hertz, P. F.; Costa, T. M. H. High-performance bio-catalyst based on β-D-galactosidase immobilized on mesoporous silica/titania/chitosan material. Food Chem. 2021, 359, 129890

Reviewer 2 Report
Review the document - microorganisms are not always written italic.
Figure 2 is oversized.
Please add the following inf: What are the various techniques for enzyme immobilization?What parameters affect the immobilization process?What are gaps?Which method is best?What are the authors recommendations? What are the specific characteristics of galactosidase improved after immobilization?
Future recommendations are a critical part of the review. It should be provided.
Being a review paper, more references need to be provided but make sure it should be recent and relevant.
Author Response
Dear reviewer,
many thanks for your suggestions.
Below you can find the response to your questions.
The corrections in the revised manuscript are made in blu font.
Best regards
Katia Liburdi
Comments (C) and response (R)
- Review the document - microorganisms are not always written in italic.
R: the document has been checked, all microorganisms' names are in italic.
C: Figure 2 is oversized.
R: The dimension of Fig. 2 is now adjusted.
C: Please add the following inf: What are the various techniques for enzyme immobilization?What parameters affect the immobilization process?What are gaps?Which method is best?What are the authors recommendations? What are the specific characteristics of galactosidase improved after immobilization?
R: two immobilization techniques are reported in Paragraph 3, covalent binding on different carriers is reported at lines 216-246 as well as the immobilization by ionic interactions at lines 247-258. As suggested, now at line 235 the oriented immobilization is referred to as the covalent bonding.
The gap with the best immobilization technique and the advantages relative to the use of immobilized b- galactosidase is now reported at lines 259-264
C: Future recommendations are a critical part of the review. It should be provided.
R: I’ve added a brief discussion about the future recommendation in the last lines of the “Conclusions” paragraph (lines 311-315).
C: Being a review paper, more references need to be provided but make sure it should be recent and relevant.
R: more recent references are now reported:
Abdul Manas, N. H.; Md. Illias, R.; Mahadi, N. M. Strategy in manipulating transglycosylation activity of glycosyl hydrolase for oligosaccharide production. Crit. Rev. Biotech. 2018, 38(2), 272-293
de Albuquerque, T. L.; de Sousa, M.; Silva, N. C. G.; Neto, C. A. C. G.; Gonçalves, L. R. B.; Fernandez-Lafuente, R.; Rocha, M. V. P. β-Galactosidase from Kluyveromyces lactis: Characterization, production, immobilization and applications-A review. Int. J. Biol. Macromol. 2021, 191, 881-898.
Fischer, C., & Kleinschmidt, T. (2018). Synthesis of galactooligosaccharides in milk and whey: a review. Comp. Rev. Food Sci. Safe. 2018, 17(3), 678-697.
Huang, J.; Zhu, S.; Zhao, L.; Chen, L.; Du, M.; Zhang, C.; Yang, S. T. A novel β-galactosidase from Klebsiella oxytoca ZJUH1705 for efficient production of galacto-oligosaccharides from lactose. App. Microb.Biotech. 2020, 104(14), 6161-6172.
Jenab, E.; Omidghane, M.; Mussone, P.; Armada, D. H.; Cartmell, J.; Montemagno, C. Enzymatic conversion of lactose into galacto-oligosaccharides: The effect of process parameters, kinetics, foam architecture, and product characterization. J. Food Eng. 2018, 222, 63-72.
Movahedpour, A.; Ahmadi, N.; Ghalamfarsa, F.; Ghesmati, Z.; Khalifeh, M.; Maleksabet, A.; ... et al. Savardashtaki, A. (). β‐Galactosidase: From its source and applications to its recombinant form. Biotech. Appl. Biochem. 2021
Neto, C. A. C. G.; Silva, N. C. G.; de Oliveira Costa, T.; de Albuquerque, T. L.; Gonçalves, L. R. B.; Fernandez-Lafuente, R.; Ro-cha, M. V. P. The β-galactosidase immobilization protocol determines its performance as catalysts in the kinetically con-trolled synthesis of lactulose. Int. J. Biol. Macromol. 2021, 176, 468-478.
Park, A.-R.; Oh, D.-K. Galacto-oligosaccharide production using microbial β-galactosidase: current state and
Ricardi, N. C.; Arenas, L. T.; Benvenutti, E. V.; Hinrichs, R.; Flores, E. E. E.; Hertz, P. F.; Costa, T. M. H. High performance bio-catalyst based on β-D-galactosidase immobilized on mesoporous silica/titania/chitosan material. Food Chem. 2021, 359, 129890

Reviewer 3 Report
The manuscript describes the factors affecting production of GOS using lactase for lactose-free milk application and the immobilization of lactase. The manuscript can be improved according to these comments:
- The title does not specifically reflect the content and aim of the review, thus needs major change. (i.e. simultaneous reactions and challenges are not discussed in the manuscript)
- In Abstract, the authors wrote the aim is to identify novel microbial source... This is a little bit confusing as it does not reflect the core content or highlight of this manuscript.
- The core of discussion is focusing on GOS production, and the highlighted application is to produce FLM. However it is not clear whether FLM produced from the enzymatic reaction of lactase contains high GOS or glucose and galactose instead.
- The process for FLM production, yield of GOS/glucose/galactose should be discussed and these data are lacking. Put one table that summarize these data - would be very helpful to readers.
- Line 95-101 needs major revision to really reflect content (i.e. no study on GOS production in LFM products are cited, structure of β-galactosidase is not discussed, GOS structure is not discussed, 'at industrial level' - but very few example of industry process given.)
- Line 151-155 - discuss more in detail about pH effects, refer https://doi.org/10.1080/07388551.2017.1339664)
- Line 162-163 - "...it could provide better control of 162
the solubility of lactose as a substrate, while inhibiting contamination. " explain more about the statement, it is not clear. - Line 156 - What is the relationship between enzyme source and lactose conversion? It is more related to the enzymatic reaction conditions i.e. lactose concentration as acceptor. Please relate this with the thermophilic properties of microorganism source.
- Line 168 - give examples of product inhibition problem
- Line 175 and above - more recent references should be included as this part discusses the recent advances. Only one 2021 paper was cited and too many old references.
- Line 195 - report the yield of GOS from the process in The Centrale del Latte of Milan, Italy.
- Line 197 - "However, the transgalactsylation efficiency of immobilized enzymes has been poorly investigated. " - what is the base of the statement?
- Line 199 - why specific emphasis on A. oryzae?
- Line 201-207 - are you refering to [38] or [34] for this finding?
- Line 201-212 - how the geometry and structure affects the reaction?
- Line 217 - explain more about heterofunction
- Line 223 and other - why specific emphasis is given to A. oryzae? What about lactase from other organism?
- Paragraph construction needs to be revised, some paragraphs are very short. These paragraphs that explain the same point can be joined as one paragraph.
- Be consistent in the terms used (e.g. LFM or FLM?, β-galactosidase or β galactosidase or βG?)
Author Response
Dear reviewer,
many thanks for your suggestions.
Below you can find the response to your questions.
The corrections in the revised manuscript are made in green font.
Best regards
Katia Liburdi
The manuscript describes the factors affecting production of GOS using lactase for lactose-free milk application and the immobilization of lactase. The manuscript can be improved according to these comments:
- C: The title does not specifically reflect the content and aim of the review, thus needs major change. (i.e. simultaneous reactions and challenges are not discussed in the manuscript)
R: The GOS are lactose-derived compounds that are produced during hydrolysis in a side reaction called transgalactosylation by the β-galactosidase (EC 3.2.1.23). This is why the word “simultaneous” is included in the title. However, I agree with the reviewer concerning the lack of relationship between the title and the content/aim of the review. For that, the title has now been changed following the reviewer’s suggestion: “ Galacto-oligosaccharide (GOS) synthesis during enzymatic lactose-free milk production: state of the art and emerging opportunities”
- C: In Abstract, the authors wrote the aim is to identify novel microbial source... This is a little bit confusing as it does not reflect the core content or highlight of this manuscript. The core of discussion is focusing on GOS production, and the highlighted application is to produce FLM.
R: effectively, the sentence in line 18 is not clear. “Current research” is not referred to the present manuscript but to the interest of the research field community. Thus, the phrase has been changed “In this field, different works aims to …” try to make the concept clear.
- C: However it is not clear whether FLM produced from the enzymatic reaction of lactase contains high GOS or glucose and galactose instead. The process for FLM production, yield of GOS/glucose/galactose should be discussed and these data are lacking. Put one table that summarize these data - would be very helpful to readers.
R: As suggested by the reviewer, the mechanisms of the simultaneous hydrolysis and trans galactosylation performed by b- galactosidase has now been extensively addressed in the manuscript at lines 115-133. Regarding the oligosaccharides production applied to the lactose-free milk production (with the relative yield of GOS/glucose/galactose) most of the research has been carried out using buffered lactose solution. However, this aspect has been substantially treated by Fischer [18] in a recent publication. For this reason, I have included and discussed this reference (Line 138-141).
- C: line 95-101 needs major revision to really reflect content (i.e. no study on GOS production in LFM products are cited, structure of β-galactosidase is not discussed, GOS structure is not discussed, 'at industrial level' - but very few example of industry process given.)
R: lines 109-113: the sentence has been rewritten following the reviewer’s suggestion.
- C: Line 151-155 - discuss more in detail about pH effects, refer https://doi.org/10.1080/07388551.2017.1339664)
R: Lines 177-180, the discussion above the pH effect as been inserted using the mentioned reference
- C: Line 162-163 - "...it could provide better control of 162 the solubility of lactose as a substrate, while inhibiting contamination. " explain more about the statement, it is not clear.
R: lines 184-186: the sentence has been rewritten to be more clear.
- C: Line 156 - What is the relationship between enzyme source and lactose conversion? It is more related to the enzymatic reaction conditions i.e. lactose concentration as acceptor. Please relate this with the thermophilic properties of microorganism source
R: Lines 97-104: As also suggested by other reviewers this consideration was made at lines 97-106.
- Line 175 and above - more recent references should be included as this part discusses the recent advances. Only one 2021 paper was cited and too many old references.
R: different recent papers of 2021 are now inserted:
de Albuquerque, T. L.; de Sousa, M.; Silva, N. C. G.; Neto, C. A. C. G.; Gonçalves, L. R. B.; Fernandez-Lafuente, R.; Rocha, M. V. P. β-Galactosidase from Kluyveromyces lactis: Characterization, production, immobilization and applications-A review. Int. J. Biol. Macromol. 2021, 191, 881-898.
Movahedpour, A.; Ahmadi, N.; Ghalamfarsa, F.; Ghesmati, Z.; Khalifeh, M.; Maleksabet, A.; ... et al. Savardashtaki, A. (). β‐Galactosidase: From its source and applications to its recombinant form. Biotech. Appl. Biochem. 2021
Neto, C. A. C. G.; Silva, N. C. G.; de Oliveira Costa, T.; de Albuquerque, T. L.; Gonçalves, L. R. B.; Fernandez-Lafuente, R.; Rocha, M. V. P. The β-galactosidase immobilization protocol determines its performance as catalysts in the kinetically controlled synthesis of lactulose. Int. J. Biol. Macromol. 2021, 176, 468-478.
Ricardi, N. C.; Arenas, L. T.; Benvenutti, E. V.; Hinrichs, R.; Flores, E. E. E.; Hertz, P. F.; Costa, T. M. H. High performance biocatalyst based on β-D-galactosidase immobilized on mesoporous silica/titania/chitosan material. Food Chem. 2021, 359, 129890.
- Line 195 - report the yield of GOS from the process in The Centrale del Latte of Milan, Italy.
R: the reference is not related to the GOS production but to the lactose hydrolysis. This is now explicated at line 220.
- Line 197 - "However, the transgalactsylation efficiency of immobilized enzymes has been poorly investigated. " - what is the base of the statement?
R: The statement has been deleted.
- Line 199 - why specific emphasis on A. oryzae?
R: because this microbial source is employed to produce most of the beta-galactosidase commercial preparation.
- Line 201-207 - are you refering to [38] or [34] for this finding?
R: all the references have been checked and adjusted
- Line 201-212 - how the geometry and structure affects the reaction?
R: as highlighted by the reviewer, the sentence at lines 224-226 is not clear and it is now rewritten (lines 232-234)
- Line 217 - explain more about heterofunction
R: The concept has been explained in lines 240-242.
- Line 223 and other - why specific emphasis is given to A. oryzae? What about lactase from other organism?
R: The frequent use of this enzyme in commercial preparations makes it more interesting for application in immobilized form.
- Paragraph construction needs to be revised, some paragraphs are very short. These paragraphs that explain the same point can be joined as one paragraph.
R: Sorry but without the indication of lines I’m not able to identify the correction to be made
- Be consistent in the terms used (e.g. LFM or FLM?, β-galactosidase or β galactosidase or βG?)
R: the use of the acronym has now been homogenous by replacing FLM with LFM and deleting the βG.
